# The deubiquitylase Ubp15 couples transcription to mRNA export

**Fanny Eyboulet[1,2], Célia Jeronimo[1], Jacques Côté[2], François Robert[1,3]\***

[1]Institut de recherches cliniques de Montréal, Montréal, Canada; [2]St-Patrick Research Group in Basic Oncology, Laval University Cancer Research Center, Axe Oncologie du Centre de Recherche du CHU de Québec-Université Laval, Québec City, Canada; [3]Département de Médecine, Faculté de Médecine, Université de Montréal, Montréal, Canada

**Abstract** Nuclear export of messenger RNAs (mRNAs) is intimately coupled to their synthesis. pre-mRNAs assemble into dynamic ribonucleoparticles as they are being transcribed, processed, and exported. The role of ubiquitylation in this process is increasingly recognized but, while a few E3 ligases have been shown to regulate nuclear export, evidence for deubiquitylases is currently lacking. Here we identified deubiquitylase Ubp15 as a regulator of nuclear export in *Saccharomyces cerevisiae*. Ubp15 interacts with both RNA polymerase II and the nuclear pore complex, and its deletion reverts the nuclear export defect of E3 ligase Rsp5 mutants. The deletion of *UBP15* leads to hyper-ubiquitylation of the main nuclear export receptor Mex67 and affects its association with THO, a complex coupling transcription to mRNA processing and involved in the recruitment of mRNA export factors to nascent transcripts. Collectively, our data support a role for Ubp15 in coupling transcription to mRNA export.

**\*For correspondence:**
francois.robert@ircm.qc.ca

**Competing interests:** The authors declare that no competing interests exist.

## Introduction

In eukaryotes, RNA polymerase II (RNAPII) is the enzyme responsible for the transcription of all protein-coding genes, as well as several noncoding RNAs. Rpb1, the largest subunit of RNAPII, contains a C-terminal domain (CTD) composed of 26 to 52 repetitions (in yeast and mammals, respectively) of the heptapeptide $Y_1S_2P_3T_4S_5P_6S_7$ (*Chapman et al., 2008*). The CTD is highly conserved and essential for the viability in all organisms. During transcription, the CTD is extensively and dynamically phosphorylated to coordinate the binding of proteins involved in the different steps of transcription and to couple transcription to mRNA processing (*Corden, 2013*; *Eick and Geyer, 2013*; *Harlen and Churchman, 2017*; *Jeronimo et al., 2013*; *Jeronimo et al., 2016*).

During their synthesis, processing and export, pre-mRNAs and mature mRNAs are packaged with RNA-binding proteins to form dynamic ribonucleoprotein particles (mRNPs) (*Mitchell and Parker, 2014*; *Singh et al., 2015*; *Tutucci and Stutz, 2011*). The mRNA export machinery is highly conserved from yeast to humans and mainly depends on the export factor Mex67-Mtr2 (TAP-NXF1 in mammals) (*Niño et al., 2013*; *Scott et al., 2019*; *Wende et al., 2019*). Mex67, in complex with Mtr2, mediates transport through the nuclear pore complex (NPC) via interactions with several FG-enriched nucleoporins (Nups) (*Santos-Rosa et al., 1998*; *Strässer et al., 2000a*; *Terry et al., 2007*). Mex67 does interact with mRNAs via different mRNA-binding adaptors. Yra1 (ALY/REF in mammals) is the first adaptor to intervein by facilitating the recruitment of Mex67 to the mRNP (*Strässer and Hurt, 2000b*; *Zenklusen et al., 2001*) while Nab2 has been reported to form a complex with Mex67 and Yra1 (*Batisse et al., 2009*; *Iglesias et al., 2010*). Interestingly, an excess of Nab2 can bypass the loss of Yra1 (*Iglesias et al., 2010*). Npl3, an RNA-binding protein, also directly interacts with Mex67 and participates in mRNA nuclear export (*Lee et al., 1996*).

The assembly of pre-mRNAs into mRNPs is functionally and physically coupled to transcription. Indeed, TREX, a protein complex composed of THO (Hpr1, Tho2, Mft1, and Thp2), the Mex67 adapters Yra1 and the DEAD-box ATPase Sub2 (*Strässer et al., 2002*), is recruited to the transcribing RNAPII and helps in the assembly of an export competent mRNP. THO recruitment to active chromatin is facilitated by the Tho2 subunit (*Peña et al., 2012*). Then, Hpr1 binds Sub2 on nascent transcripts and is implicated in the early recruitment of Mex67, an interaction mediated by a ubiquitin-dependent process (*Gwizdek et al., 2005*; *Gwizdek et al., 2006*; *Zenklusen et al., 2002*). Yra1 is also co-transcriptionally recruited, notably by the cap binding complex (CBC) (*Cheng et al., 2006*; *Sen et al., 2019*; *Viphakone et al., 2019*) and transferred to Sub2 (*Johnson et al., 2009*; *Johnson et al., 2011*). Later, TREX-2 (Sac3, Thp1, Sus1, Cdc31, and Sem1) interacts with the nuclear side of the NPC and participates in the coordination between transcription and mRNA export (*Cabal et al., 2006*; *Fischer et al., 2004*; *Jani et al., 2009*; *Rodríguez-Navarro et al., 2004*).

In contrast to Yra1, which is removed from mRNP complexes before exiting the nucleus (*Lund and Guthrie, 2005*), Nab2 and Npl3 translocate together with the mRNP and are released at the cytoplasmic face of the NPC (*Tran et al., 2007*). Once the mRNA reaches the cytoplasm, Nup159 and Nup42 NPC subunits recruit two essential mRNA export factors, the DEAD-box ATPase Dbp5 and its ATPase activator Gle1, which are in charge of remodeling and disassembling the mRNPs emerging from the NPC (*Adams et al., 2017*; *Hodge et al., 2011*; *Lund and Guthrie, 2005*; *Weirich et al., 2006*). Disassembly of the mRNP in the cytoplasm prevents its return to the nucleus, resulting in unidirectional mRNA translocation. In addition, the release of mRNA export factors allow them to return to the nucleus where they can function in additional rounds of mRNA export (*Stewart, 2007*). Recently, it was shown that Dbp5 does not affect the interaction of Mex67 with the NPC. Instead, it was proposed that Mex67 does not translocate to the cytoplasmic face of the NPC with the mRNPs but rather functions as a mobile nucleoporin, facilitating their translocation to the cytoplasm (*Derrer et al., 2019*).

Protein ubiquitylation plays very important roles in the control of numerous cellular pathways. In gene expression, ubiquitin has been shown to regulate the activity and turnover of several key transcription factors. The Rsp5 ubiquitin ligase was shown to target RNAPII for degradation during DNA damage (*Beaudenon et al., 1999*; *Huibregtse et al., 1997*; *Wu et al., 2001*). More recently, ubiquitylation of RNAPII in human cells was shown to play a role in transcription-coupled repair (*Nakazawa et al., 2020*; *Tufegdžić Vidaković et al., 2020*). RNAPII was also shown to be ubiquitylated in vivo by the ubiquitin ligase Asr1 (*Daulny et al., 2008*). In this case, the ubiquitylation occurs on transcribed genes and leads to the dissociation of two RNAPII subunits (Rpb4/7), a process involved in the silencing of subtelomeric genes (*McCann et al., 2016*). Ubiquitylation is also involved in mRNA export (*Babour et al., 2012*). The THO component Hpr1 is poly-ubiquitylated by Rsp5, which facilitates the co-transcriptional recruitment of the mRNA export factor Mex67, via its ubiquitin-associated (UBA) domain (*Gwizdek et al., 2005*; *Gwizdek et al., 2006*). The ubiquitin E3 ligase Tom1 is required for ubiquitylation of the Mex67 adaptor Yra1, which promotes its dissociation from mRNP complexes before export to the cytoplasm (*Iglesias et al., 2010*). H2B ubiquitylation, a histone modification deposited on genes co-transcriptionally, facilitates the assembly of Mex67, Yra1, Nab2, and Npl3 into mRNPs (*Vitaliano-Prunier et al., 2012*). Interestingly, while H2B ubiquitylation mediates the assembly of Npl3 directly, its effect on Mex67, Yra1, Nab2 assembly onto mRNPs involves the ubiquitylation of Swd2, a subunit of the cleavage and polyadenylation factor (CPF), connecting chromatin to transcriptional termination and mRNA export (*Vitaliano-Prunier et al., 2012*). Moreover, a systematic analysis of NPC ubiquitylation was conducted in yeast and showed that more than 50% of Nups can be ubiquitylated (*Hayakawa et al., 2012*). As this modification does not influence Nups localization within the NPC, it is tempting to speculate that it might regulate the interaction with the Mex67 UBA domain during mRNA export.

Here we identified the deubiquitylase Ubp15 as an RNAPII CTD-interacting protein in *Saccharomyces cerevisiae*. The association of Ubp15 with RNAPII increased in a mutant for the CTD phosphatase Fcp1, suggesting a role in transcriptional elongation. Furthermore, the deletion of *UBP15* rescued the sensitivity of transcription elongation factor mutants to 6-azauracil (6AU), an inhibitor of transcription elongation. While these experiments functionally connect Ubp15 to transcription elongation, the deletion of *UBP15* did not rescue the transcription elongation processivity defect of *dst1Δ* cells in the presence of 6AU, suggesting that the link between Ubp15 and elongation may be indirect. The deletion of *UBP15* rescued the thermosensitivity of a mutant for *RSP5*, an E3 ligase

involved in DNA damage and mRNA export. Interestingly, the deletion of *UBP15* suppressed the mRNA export defect of *rsp5* mutants. Finally, we showed that Ubp15 regulates the ubiquitylation of Mex67, which in turn regulates its interaction with THO, a complex involved in mRNA processing and export. Collectively, our data support a role for Ubp15 in coupling transcription to mRNA export.

## Results

### Ubp15 associates with RNAPII and the NPC

To identify new factors interacting with RNAPII in a CTD phosphorylation-dependent manner, we performed a proteomic analysis of RNAPII complexes, affinity-purified from wild type (WT) and *fcp1-1* cells (*Figure 1A*, *Figure 1—figure supplement 1*). Fcp1 is a major CTD phosphatase and its mutation leads to increased CTD phosphorylation at serines 2, 5, and 7 (*Bataille et al., 2012*). Quantitative analysis of the RNAPII-associated proteins identified 45 proteins differentially associated with the polymerase in *fcp1-1* cells (35 being more abundant and 10 being less abundant by at least twofold; *Figure 1B*, *Supplementary file 1*). One RNAPII-interacting protein identified in this experiment is Ubp15, a deubiquitylase known for its role in the regulation of endocytosis (*Ho et al., 2017*; *Kouranti et al., 2010*), progression into the S phase (*Álvarez et al., 2016*; *Ostapenko et al., 2015*), peroxisomal export (*Debelyy et al., 2011*), and methylmercury susceptibility (*Hwang et al., 2012*), but with no known roles in transcription. The association of Ubp15 with RNAPII is increased in the *fcp1-1* mutant, suggesting it is recruited (directly or indirectly) to the elongating polymerase via CTD phosphorylation. To confirm this interaction, we performed a reciprocal affinity purification experiment where Ubp15 was affinity-purified in WT cells and analyzed by mass spectrometry (*Figure 1C*) and western blot (*Figure 1D*). This experiment confirmed the interaction between Ubp15 and the phosphorylated form of RNAPII (*Figure 1D*) and, surprisingly, revealed the enrichment of almost the entire NPC (*Figure 1C*). Collectively, these experiments identified Ubp15 as an interactor of the phosphorylated RNAPII and the NPC.

### Ubp15 genetically interacts with elongation factors

Interestingly, the histone chaperone and elongation factor FACT (Spt16 and Pob3) is also enriched in RNAPII purified from *fcp1-1* cells (*Figure 1B*, *Figure 1—figure supplement 1B*, *Supplementary file 1*) and both FACT and Rpb1 are substrates of Ubp15 in *Schizosaccharomyces pombe* (*Beckley et al., 2015*). In addition, the association of Asr1, a ubiquitin ligase known to bind the CTD and to ubiquitylate RNAPII in the context of elongation (*Daulny et al., 2008*; *McCann et al., 2016*), is decreased in *fcp1-1* cells (*Figure 1B*, *Supplementary file 1*). These results prompted us to investigate the possible role for Ubp15 in transcription elongation.

First, we looked for sensitivity to 6-azauracil (6AU), an inhibitor of GTP biosynthesis commonly used to test for mutations that affect transcriptional elongation (*Riles et al., 2004*). While the *ubp15Δ* mutant did not show significant sensitivity to 6AU on its own, it rescued the 6AU sensitivity of several elongation factor mutants including *dst1Δ* (TFIIS), *spt4Δ* and *spt5-CTRΔ* (DSIF), *spt6-1004* and *hpr1Δ* (THO) (*Figure 2A*, *Figure 2—figure supplement 1A*). Interestingly, the 6AU sensitivity of other elongation factors such as *bur2Δ*, *ctk1Δ*, *rtf1Δ*, and *cdc73Δ* mutants was not affected by the deletion of *UBP15* (*Figure 2—figure supplement 1A*). Then, we used a Ubp15 catalytic dead mutant, *ubp15-C214A* (*Debelyy et al., 2011*), to test whether the observed genetic interactions were mediated by its catalytic activity. Double mutants *ubp15Δ/dst1Δ* and *ubp15Δ/spt4Δ* were complemented with plasmids expressing WT or C214A versions of Ubp15 and spotted on 6AU. The *ubp15-C214A* mutant, but not the WT *UBP15*, rescued the *dst1Δ* and *spt4Δ* sensitivities to 6AU (*Figure 2B*), demonstrating that the effect of Ubp15 on elongation depends on its catalytic activity.

Noteworthy, we also looked at the phenotype of *ubp15Δ* cells under several other growth conditions and confirmed previously described sensitivity to cold temperature and hydroxyurea (HU) (*Amerik et al., 2000*; *Ostapenko et al., 2015*), while elevated temperature, ultraviolet light, formamide or caffeine had no detectable effect (*Figure 2—figure supplement 1B*). Collectively, these genetic interactions confirm the previous literature on Ubp15 and demonstrate a functional link between Ubp15 deubiquitylase activity and transcription elongation.

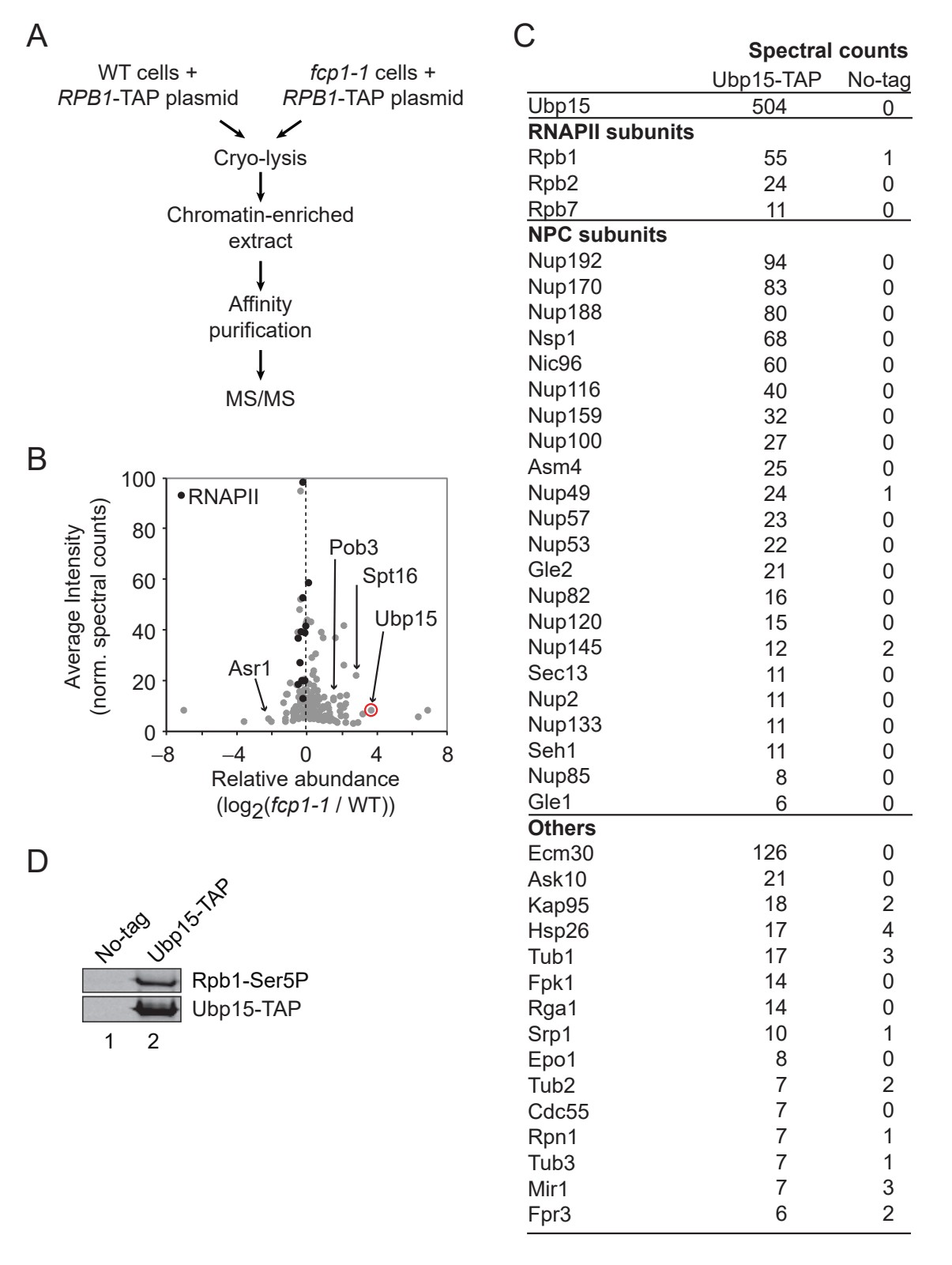

**A**

WT cells +
*RPB1*-TAP plasmid        *fcp1-1* cells +
                          *RPB1*-TAP plasmid

Cryo-lysis

Chromatin-enriched
extract

Affinity
purification

MS/MS

**B**

Average Intensity (norm. spectral counts) vs Relative abundance (log$_2$(*fcp1-1* / WT))

Labels: RNAPII, Pob3, Spt16, Ubp15, Asr1

**D**

No-tag | Ubp15-TAP

Rpb1-Ser5P
Ubp15-TAP

1    2

**C**

| | Spectral counts | |
| --- | --- | --- |
| | Ubp15-TAP | No-tag |
| Ubp15 | 504 | 0 |
| **RNAPII subunits** | | |
| Rpb1 | 55 | 1 |
| Rpb2 | 24 | 0 |
| Rpb7 | 11 | 0 |
| **NPC subunits** | | |
| Nup192 | 94 | 0 |
| Nup170 | 83 | 0 |
| Nup188 | 80 | 0 |
| Nsp1 | 68 | 0 |
| Nic96 | 60 | 0 |
| Nup116 | 40 | 0 |
| Nup159 | 32 | 0 |
| Nup100 | 27 | 0 |
| Asm4 | 25 | 0 |
| Nup49 | 24 | 1 |
| Nup57 | 23 | 0 |
| Nup53 | 22 | 0 |
| Gle2 | 21 | 0 |
| Nup82 | 16 | 0 |
| Nup120 | 15 | 0 |
| Nup145 | 12 | 2 |
| Sec13 | 11 | 0 |
| Nup2 | 11 | 0 |
| Nup133 | 11 | 0 |
| Seh1 | 11 | 0 |
| Nup85 | 8 | 0 |
| Gle1 | 6 | 0 |
| **Others** | | |
| Ecm30 | 126 | 0 |
| Ask10 | 21 | 0 |
| Kap95 | 18 | 2 |
| Hsp26 | 17 | 4 |
| Tub1 | 17 | 3 |
| Fpk1 | 14 | 0 |
| Rga1 | 14 | 0 |
| Srp1 | 10 | 1 |
| Epo1 | 8 | 0 |
| Tub2 | 7 | 2 |
| Cdc55 | 7 | 0 |
| Rpn1 | 7 | 1 |
| Tub3 | 7 | 1 |
| Mir1 | 7 | 3 |
| Fpr3 | 6 | 2 |

**Figure 1.** Ubp15 interacts with the phosphorylated RNAPII and with the NPC. (**A**) A schematic representation of the proteomic experiment shown in panel B. (**B**) Volcano plot showing the average intensity versus relative abundance (log$_2$) of proteins identified in RNAPII complexes purified from *fcp1-1* versus WT cells using spectral counts as a proxy for relative protein abundance. RNAPII subunits are shown in black and other proteins of interest are indicated. Ubp15 is circled in red. In the interest of clarity, the maximum value of the y-axis was set at 100, resulting in Rpb1 and Rpb2 not appearing

*Figure 1 continued on next page*

*Figure 1 continued*

on the graph. See *Supplementary file 1* for the complete list of values. (C) Spectral counts of the proteins identified in a TAP-tag purification of Ubp15. (D) Western blot confirming that Ubp15 is associated with a phosphorylated form of RNAPII (Rpb1-Ser5P).

The online version of this article includes the following figure supplement(s) for figure 1:

**Figure supplement 1.** Rpb1-TAP complexes purified from WT and fcp1-1 cells.

## Ubp15 does not regulate RNAPII processivity

The suppression of the 6AU sensitivity of several elongation factors is consistent with a role for Ubp15 in transcription elongation. In *S. cerevisiae*, elongation factors affect the processivity of RNA-PII (i.e. the capacity of the polymerase to reach the end of the gene) but have no measurable impact on the elongation rate (*Mason and Struhl, 2005*). We, therefore, tested the effect of *UBP15* deletion on RNAPII processivity in WT and *dst1Δ* cells. We first compared RNAPII occupancy at the beginning and the end of *YLR454W* (a ~ 8 kb-long gene) by chromatin immunoprecipitation (ChIP) followed by quantitative PCR (qPCR) as described before (*Mason and Struhl, 2003*; *Mason and Struhl, 2005*; *Schwabish and Struhl, 2004*; *Strässer et al., 2002*). Surprisingly, we found that the deletion of *UBP15* did not rescue the processivity defect of *dst1Δ* cells in the presence of 6AU (*Figure 3A*). To extend this analysis to the whole genome, we performed ChIP followed by hybridization on tiling microarrays (ChIP-chip) experiments in WT, *ubp15Δ*, *dst1Δ,* and *ubp15Δ/dst1Δ* cells and mapped average RNAPII occupancy over transcribed genes (*Figure 3B*, *Figure 3—figure supplement 1*). These experiments confirmed the processivity defect of *dst1Δ* cells but, consistently with our ChIP-qPCR analysis of the *YLR454W* gene (*Figure 3A*), revealed no effect of *UBP15* deletion on RNAPII processivity in WT or *dst1Δ* cells. This result was somewhat unexpected given the genetic interactions described in *Figure 2* and suggests that *UBP15* deletion may rescue the 6AU sensitivity of elongation factor mutants indirectly. Alternatively, Ubp15 may affect elongation rate, a parameter not investigated here, but this appears unlikely since no elongation factor has been shown to affect this parameter in yeast (*Mason and Struhl, 2005*). One attractive possibility, given the interaction between Ubp15 and the NPC, is that *UBP15* deletion affects a post-transcriptional process in a way that compensates for elongation defects of elongation factor mutants.

## Ubp15 controls nuclear polyA RNA accumulation

To identify a role for Ubp15 that may explain its indirect elongation phenotype, we looked for an E3 ligase that would oppose its function. Through testing several candidates (*Figure 4A*, *Figure 4—figure supplement 1A*), we noticed that a mutant of the E3 ligase Rsp5 (*rsp5-1*) exacerbated the 6AU sensitivity of *dst1Δ* cells (*Figure 4A*). This effect of *rsp5-1* on the *dst1Δ* 6AU sensitivity is opposite to that of *ubp15Δ* (*Figure 2A*) suggesting that Rsp5 and Ubp15 may oppose each other in regulating a process genetically connected to transcription elongation. Furthermore, the deletion of *UBP15* can partially rescue the 6AU sensitivity of *rsp5-1/dst1Δ* mutant (*Figure 4A*). The antagonistic relationship between Rsp5 and Ubp15 is also supported by the fact that the deletion of *UBP15* rescued the thermosensitivity phenotype of *rsp5-ts* mutants (*Figure 4B* and *Figure 4—figure supplement 1B*). Importantly, the rescue of *rsp5-1* thermosensitivity was also observed with the *ubp15-C214A* mutant indicating it is dependent on the catalytic activity of Ubp15 (*Figure 4C*). Because we found that overexpression of Rps5-1 can rescue the *rsp5-1* thermosensitivity phenotype (*Figure 4—figure supplement 1C*), we next tested whether Rsp5 protein levels were affected by the deletion of *UBP15*. As shown in *Figure 4—figure supplement 1D*, Rsp5 protein levels in *rsp5-1* cells are actually decreased upon deletion of *UBP15*. This result suggests that *UBP15* deletion suppresses the *rsp5-1* thermosensitivity phenotype by other mechanisms, as explored further below.

Rsp5 is an E3 ligase involved in DNA damage (*Beaudenon et al., 1999*) and mRNA export (*Rodriguez et al., 2003*). At high temperatures, the *rsp5-1* mutant accumulates mRNA in the nucleus (*Rodriguez et al., 2003*). We, therefore, compared polyA RNA localization in *rsp5-1* and *ubp15Δ/rsp5-1* cells by RNA fluorescent in situ hybridization (FISH) using a Cy5-labeled oligo-dT$_{45}$ probe. Interestingly, these experiments showed that the deletion of *UBP15* suppressed the mRNA export defect of the *rsp5-1* mutant (*Figure 4D*). Indeed, the deletion of *UBP15* reduced the number of cells retaining polyA RNAs in their nucleus from around 75% (172/222) to less than 5% (11/227). This

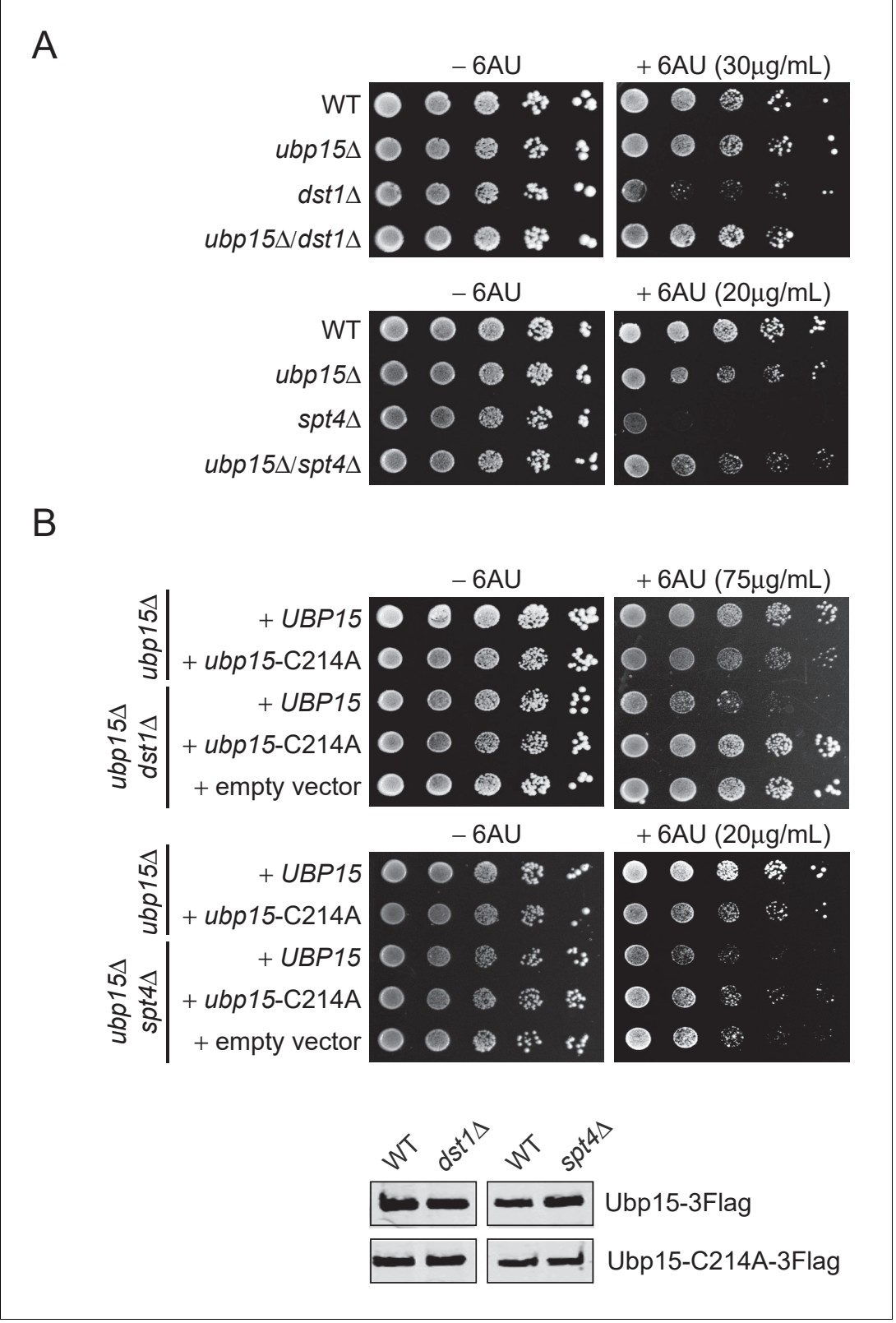

**Figure 2.** The deletion of *UBP15* suppresses the 6AU sensitivity of *dst1Δ* and *spt4Δ* cells. (**A**) Serial-dilution growth assays assessing the 6AU sensitivity of WT, *dst1Δ,* and *spt4Δ* cells, alone and in combination with *ubp15Δ*. The indicated yeast strains were grown to saturation in YNB medium lacking uracil (−URA), washed, resuspended at the same density in water, serially diluted (fivefold series), and spotted on –URA in the absence or presence of

*Figure 2 continued on next page*

*Figure 2 continued*

6AU as indicated. Plates were incubated at 30°C for 3 days. (**B**) Top: Serial-dilution growth assays assessing the requirement of the catalytic activity of Ubp15 for its effect on the 6AU sensitivity of WT, *dst1Δ,* and *spt4Δ* cells. Strains were deleted for *UBP15, DST1,* and *SPT4,* alone or in combinations, and transformed with empty vector, plasmids expressing Ubp15-3Flag (*UBP15*) or catalytic dead Ubp15-C214A-3Flag (*ubp15*-C214A) and spotted on −URA lacking histidine (−URA /−HIS) in the absence or presence of 6AU as indicated. Bottom: Equal expression levels of WT and catalytic dead versions of Ubp15-3Flag in WT, *dst1Δ*, and *spt4Δ* cells were confirmed by western blot. Note that the 6AU concentration varies and has been optimized for each mutant.

The online version of this article includes the following figure supplement(s) for figure 2:

**Figure supplement 1.** UBP15 deletion phenotypes and genetic interactions.

---

suppression of mRNA export defect likely explains how *UBP15* deletion restored the viability of *rsp5* mutants and clearly establishes a role for Ubp15 in mRNA export.

## Ubp15 and Rsp5 have opposite effects on Mex67 ubiquitylation

To identify potential substrates for Ubp15 in the mRNA export pathway, we measured ubiquitylation levels of Mex67 (*Figure 5B,C*), six NPC components (Nup82, Nup133, Nup57, Nup120, Nup145, and Nup159) (*Figure 5—figure supplement 1A*) and four nuclear export factors (Hpr1, Nab2, Npl3,

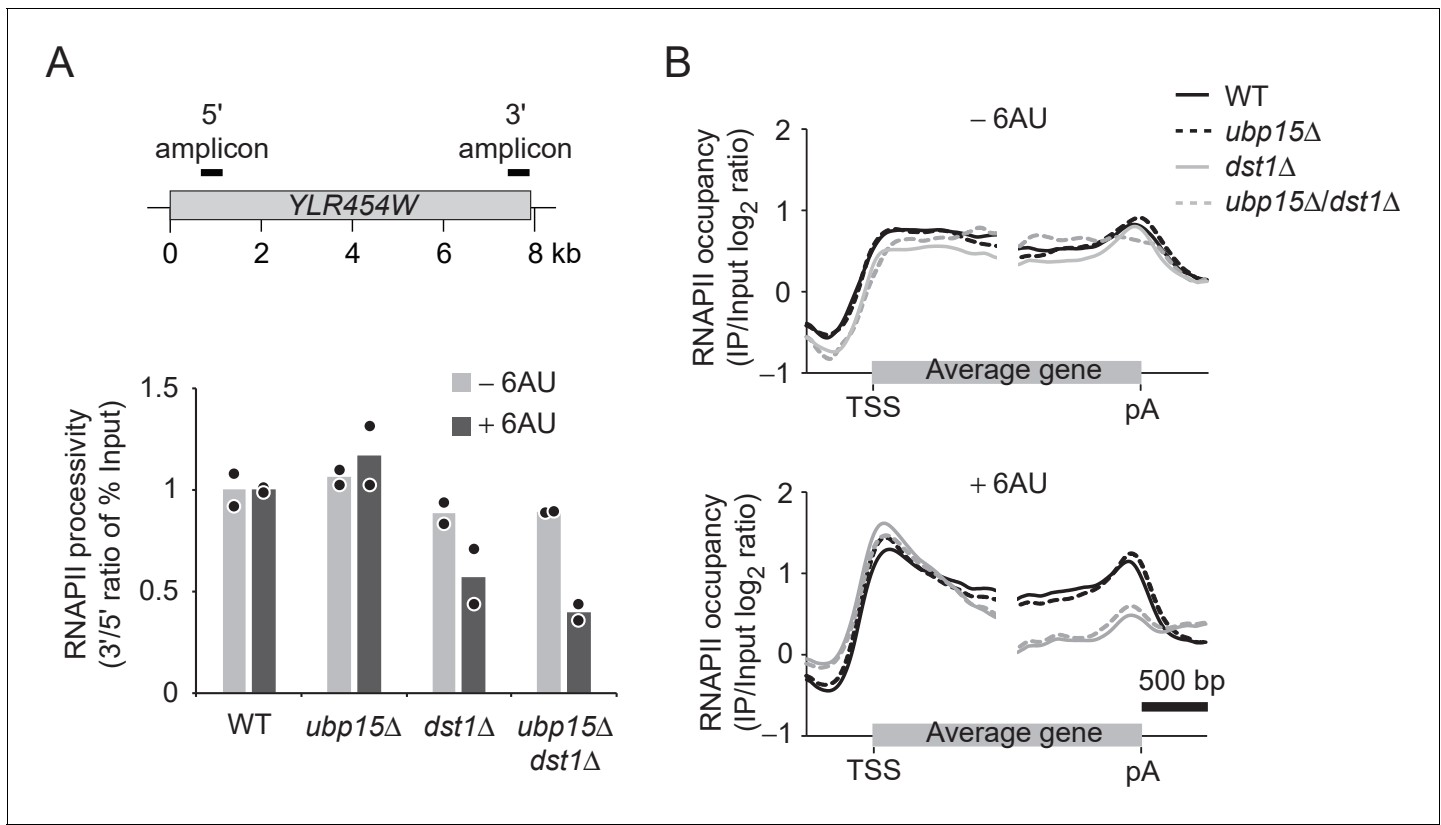

**Figure 3.** *UBP15* deletion does not rescue RNAPII processivity in *dst1Δ* cells. (**A**) RNAPII processivity, defined as the ratio of the % of Input detected in the 3′ amplicon divided by the % of Input detected in the 5′ amplicon, after 30 min treatment with 6AU (dark gray) and absence of 6AU (light gray), as determined by ChIP-qPCR. Experiments were performed in two biological replicates. Bars show the average and circles show individual replicates. The position of PCR amplicons over the *YLR454W* gene used for the qPCR is indicated on the sketch above the graphs. (**B**) Aggregate profiles of RNAPII (Rpb3) occupancy over highly expressed yeast genes longer than 1 kb (n = 234) as determined by ChIP-chip after 1 hr treatment with 6AU. TSS, transcription start site; pA, polyadenylation site.

The online version of this article includes the following figure supplement(s) for figure 3:

**Figure supplement 1.** Violin plot showing the RNAPII processivity, as determined by the log$_2$ ratio of RNAPII (Rpb3) occupancy in the last 300 bp versus the first 300 bp of each gene, as determined by ChIP-chip.

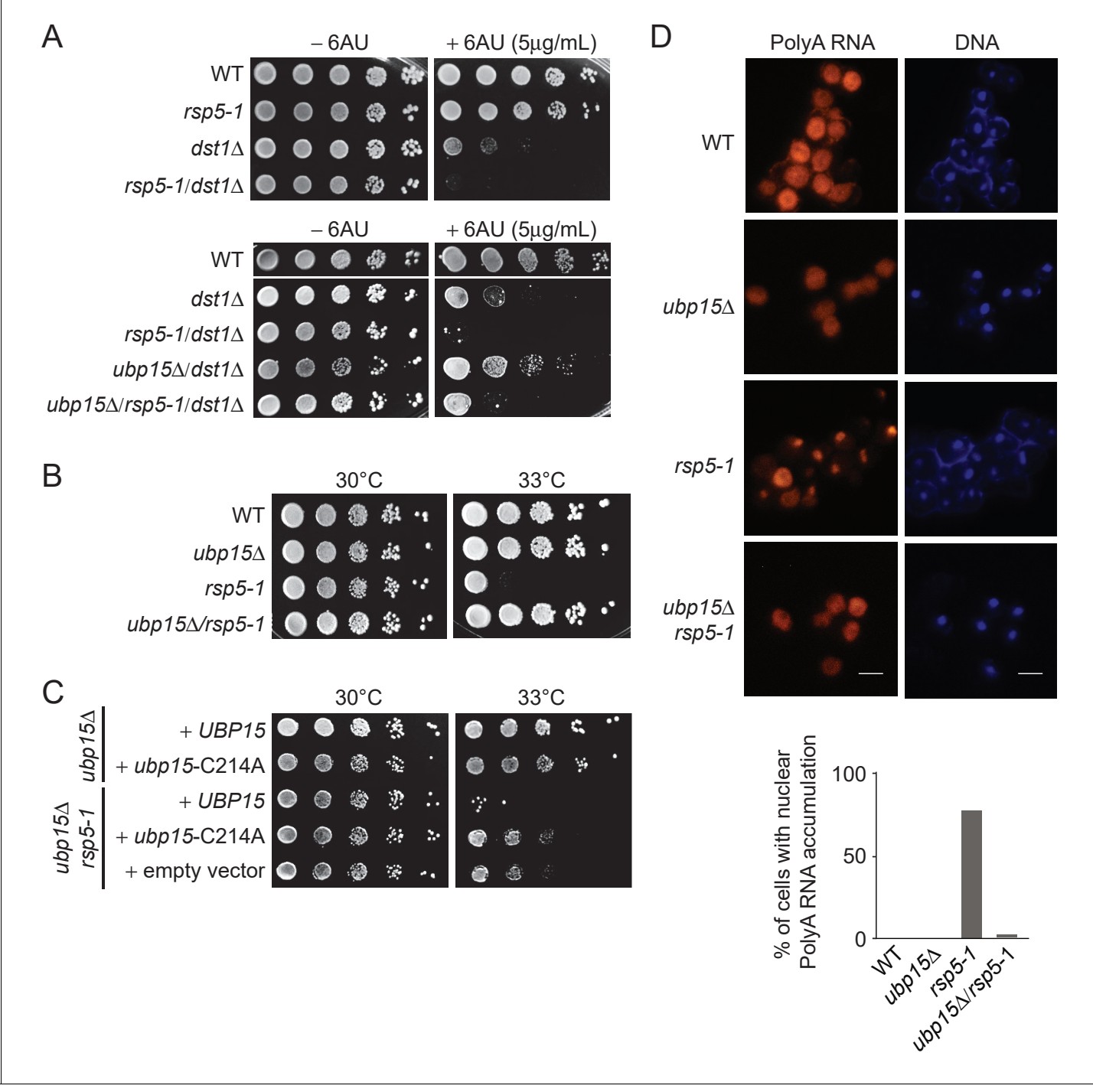

**Figure 4.** Deletion of *UBP15* rescues phenotypes of *rsp5-1* mutants. (A) Serial-dilution growth assays assessing the sensitivity of *rsp5-1* cells, alone or in combination with *dst1Δ* and or *ubp15Δ*, to 6AU. The indicated yeast strains were grown to saturation in YNB medium lacking uracil (−URA), washed, resuspended at the same density in water, serially diluted (fivefold series), and spotted on −URA in the absence or presence of 6AU as indicated. Plates were incubated at 30℃ for 3 days. (B) Serial-dilution growth assays assessing the effect of *UBP15* deletion on the viability of *rsp5-1* cells at 33℃. The indicated yeast strains were grown to saturation in YPD, washed, resuspended at the same density in water, serially diluted (fivefold series), and spotted on YPD. Plates were incubated for 3 days at 30℃ or 33℃ as indicated. (C) Serial-dilution growth assays assessing the contribution of the catalytic activity of Ubp15 to the genetic interaction between *RSP5* and *UBP15* shown in panel B. Strains deleted for *UBP15*, alone or in combination with the *rsp5-1* mutation, were transformed with empty vector, plasmids expressing Ubp15-3Flag (*UBP15*) or catalytic dead Ubp15-C214A-3Flag (*ubp15*-C214A), spotted on YNB medium lacking histidine (−HIS), and incubated at 30℃ or 33℃ as indicated. (D) RNA FISH experiments looking at bulk polyA RNAs in WT, *ubp15Δ*, *rsp5-1*, and *ubp15Δ/rsp5-1* cells (FY genetic background). The indicated strains were grown at 30℃ in YPD then shifted to 37℃ for 3 hr

*Figure 4 continued on next page*

*Figure 4 continued*

before being analyzed by FISH using Cy5-oligo-dT$_{45}$. DNA was stained with DAPI. Scale bar, 10 μm. The percentage of cells (from at least 200 cells in each strain) with retention of polyA RNA in the nucleus is indicated on the graphic shown at the bottom (WT: 0/200, *ubp15Δ*: 0/200, *rsp5-1*: 172/222, *ubp15Δ/rsp5-1*: 11/227).

The online version of this article includes the following figure supplement(s) for figure 4:

**Figure supplement 1.** A screen for E3 ligases genetically connected to UBP15.

and Mtr2) (*Figure 5—figure supplement 1B*) in *ubp15Δ* cells using a previously described in vivo ubiquitylation assay (*Figure 5A*; *Gwizdek et al., 2005*). These experiments identified Mex67 as a likely substrate of Ubp15 (*Figure 5B,C*). Interestingly, Mex67 ubiquitylation levels decreased in *rsp5-1* (*Figure 5B*), suggesting that Rsp5 and Ubp15 may oppose each other in the control of Mex67 ubiquitylation. To assess this possibility more directly, we tested Mex67 ubiquitylation levels in *ubp15Δ/rsp5-1* double mutant and found that the deletion of *UBP15* normalized Mex67 ubiquitylation of *rsp5-1* cells (*Figure 5C*). Collectively, these experiments established Rsp5 and Ubp15 as an E3 ligase/deubiquitylase tandem regulating Mex67 ubiquitylation.

## Ubp15 regulates the interaction of Mex67 with THO and the cap-binding complex

Our data so far show that Rsp5 and Ubp15 oppose each other in the regulation of mRNA export and ubiquitylation of Mex67. Since Mex67 is known to function as part of a large and dynamic protein interaction network, we hypothesized that its ubiquitylation may modulate its interactome. Hence, we tested whether the proteome associated with Mex67 is modified in *ubp15Δ* cells. We performed a proteomic analysis of Mex67, affinity-purified from WT and *ubp15Δ* cells (*Figure 5D*, *Figure 5—figure supplement 2*). As expected, based on previous studies (*Batisse et al., 2009*; *Gwizdek et al., 2006*; *Oeffinger et al., 2007*; *Santos-Rosa et al., 1998*; *Saroufim et al., 2015*; *Strässer and Hurt, 2000b*; *Zenklusen et al., 2001*), we found a large number of Mex67-interacting proteins including Mtr2, Nab2, Yra2, the THO complex and nearly the entire NPC (*Supplementary file 2*). Among those, only seven were differentially associated with Mex67 in *ubp15Δ* cells by at least twofold (*Figure 5D*). Interestingly, all those interactors were less associated with Mex67 in *ubp15Δ* cells, suggesting that the ubiquitylation of Mex67 negatively regulates its association with these factors. Strikingly, these proteins include all four subunits of the THO complex (Hpr1, Tho2, Mft1, and Thp2) and both subunits of the heterodimeric cap-binding complex (CBC; Sto1 and Cbc2) known to recruit THO/TREX to the 5'-cap of nascent pre-mRNAs (*Figure 5D*) in yeast and humans (*Cheng et al., 2006*; *Sen et al., 2019*; *Viphakone et al., 2019*). We noted also that several subunits of the NPC, including the nuclear basket protein Mlp1, showed a slightly increased association with Mex67 in *ubp15Δ* cells compared to WT cells (although below our significance threshold) (*Figure 5D*). Together, these results support a model where Mex67 ubiquitylation, controlled by Rsp5 and Ubp15, regulates the association of Mex67 to pre-mRNAs via interactions with THO and cap binding factors (*Figure 5E*).

## Discussion

In this study, we characterized the role of the deubiquitylase Ubp15 in mRNA export in yeast. Initially, we identified Ubp15 as an interactor of the phosphorylated RNAPII and the NPC. We then investigated for a role in transcription elongation and showed that the deletion of *UBP15* rescues the sensitivity of diverse transcription elongation factor mutants to 6AU but surprisingly does not rescue the transcription elongation processivity defect in these mutants. These results argue against a direct role of Ubp15 in transcriptional elongation and rather suggest that it regulates a post-transcriptional event in a way that indirectly suppresses the growth phenotype of elongation factor mutants. While looking for an E3 ligase that may oppose the function of Ubp15, we found that the deletion of *UBP15* can rescue the thermosensitivity of a mutant of *RSP5*. Interestingly, the deletion of *UBP15* suppresses the mRNA export defect of *rsp5* ts mutants, establishing a role for Ubp15 in mRNA export. We then showed that Ubp15 and Rsp5 control the ubiquitylation levels of Mex67 and

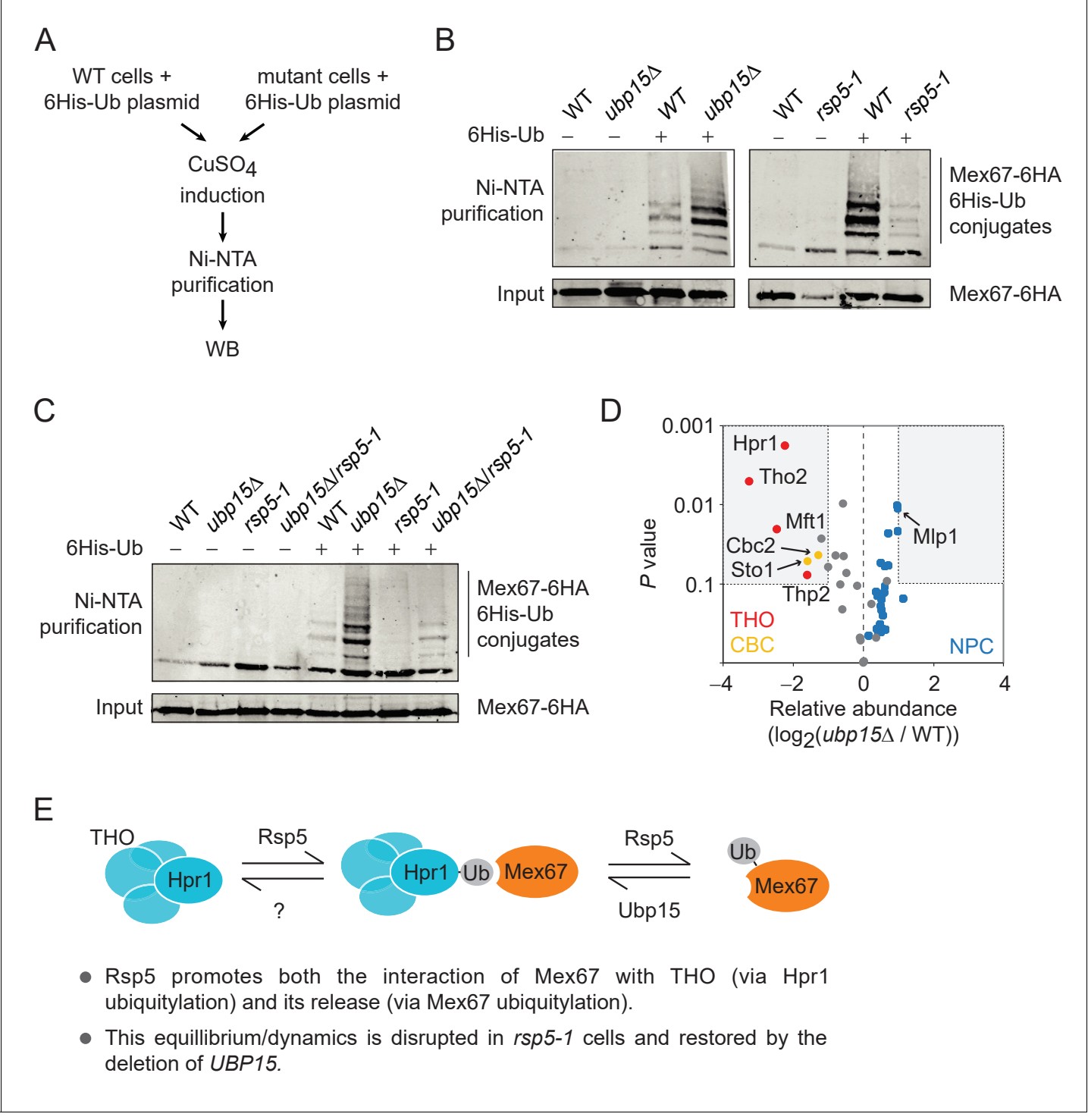

**Figure 5.** The ubiquitylation of Mex67 is regulated by Ubp15 and Rsp5. (**A**) A schematic representation of the in vivo ubiquitylation assay used in panels B and C. A plasmid expressing polyhistidine-tagged ubiquitin (6His-Ub) under the control of a copper-inducible promoter was transformed in WT and mutant cells. 6His-Ub expression was induced with copper sulfate ($CuSO_4$) and His-tagged ubiquitin-conjugated proteins were purified using Ni-NTA beads and analyzed by western blot. (**B**) Western blots for Mex67-6HA levels from His-tagged ubiquitin-conjugated protein pulldowns (Ni-NTA) and their inputs expressing (+) or not (−) 6His-Ub in WT, *ubp15Δ*, and *rsp5-1* cells. (**C**) Same as panel B but for WT, *ubp15Δ*, *rsp5-1*, and *ubp15Δ/rsp5-1* cells. (**D**) Ubp15 regulates the interaction of THO and CBC with Mex67. A volcano plot showing the significance versus the $\log_2$ fold change for the proteins identified in Mex67-3Flag purifications by MS in *ubp15Δ* versus WT cells. Gray dots show the bulk of the data and the regions for significant (p value<0.1) twofold changes are boxed. THO, CBC, and NPC subunits are labeled in red, gold, and blue, respectively. See *Supplementary file 2* for the

*Figure 5 continued on next page*

*Figure 5 continued*

complete list of values. (E) A graphical model illustrating how the ubiquitylation/deubiquitylation of Hpr1 and Mex67 by Rps5, Ubp15, and likely other deubiquitylases, may enable dynamic interactions between THO and Mex67. Both promoting and disrupting these interactions are surmise to be required for the mRNPs to mature into export-competent particles.

The online version of this article includes the following figure supplement(s) for figure 5:

**Figure supplement 1.** In vivo ubiquitylation assays testing for various Ubp15 possible substrates.

**Figure supplement 2.** A silver-stained SDS-PAGE gel showing Mex67 complexes purified from WT and *ubp15Δ* cells.

---

that Mex67 ubiquitylation negatively regulates its interaction with THO and CBC. Collectively, our data support a role for Ubp15 in coupling transcription to mRNA export.

Our initial data showed that the interaction of Ubp15 with RNAPII is increased in the mutant for the CTD phosphatase *fcp1-1*. In this mutant, RNAPII is hyper-phosphorylated, notably on its Ser2, a phosphorylation state which is predominant towards the 3'-end of genes (*Bataille et al., 2012*). mRNA processing and export factors also function near the 3'-end of genes and some even require Ser2 phosphorylation for their recruitment (*Jeronimo et al., 2013*), which could explain the link between Ubp15, RNAPII CTD phosphorylation, and mRNA export.

In principle, one possible role of Mex67 ubiquitylation/deubiquitylation by Rsp5 and Ubp15, respectively, could be to regulate its nuclear localization. However, we could not observe any effect of *ubp15Δ* or *rsp5-1* mutants on Mex67 localization by immunofluorescence (Mex67 localized at the nuclear envelope in all strains tested, data not shown). This is consistent with a recent paper from *Derrer et al., 2019* which demonstrated that Mex67 localization is restricted to the NPC. Instead, we showed that Mex67 ubiquitylation (i.e. *UBP15* deletion) negatively affects its interaction with the THO and CBC complexes. Interestingly, CBC was shown to recruit THO/TREX to the 5'-cap via interaction with Yra1 (*Cheng et al., 2006*; *Sen et al., 2019*; *Viphakone et al., 2019*). Hence, our data suggest that the assembly of Mex67 into mRNPs during transcription is counteracted by its ubiquitylation. In such a model, Ubp15, recruited by the phosphorylated RNAPII CTD, would antagonize Mex67 ubiquitylation, hence allowing Mex67 recruitment to the pre-mRNA by CBC and THO/TREX. However, we were not able to show a reduction in Mex67 occupancy on genes in *ubp15Δ* cells using ChIP (data not shown). These results, however, should be interpreted with caution since the Mex67 ChIP generated a very low signal over the background (data not shown). It is, therefore, possible that an effect on Mex67 occupancy had been missed due to technical limitations. Alternatively, Ubp15 may regulate a step in the assembly of Mex67 into mRNP that is not detectable by ChIP. For instance, Ubp15 may regulate the translocation of Mex67 from the RNAPII elongating complex onto the mRNA before its release, a step that would be more easily assayed by protein-RNA detection methods such as CRAC or PAR-CLIP.

How could ubiquitylation of Mex67 prevent its interaction with CBC and THO/TREX? Ubiquitylation of Mex67 may prevent interaction with CBC and THO/TREX by a simple sterical block or, –since Mex67 contains a UBA domain– Mex67 ubiquitylation may trigger an intramolecular interaction between the conjugated ubiquitin and the UBA domain, creating a folded closed conformation. In line with such a scenario, the UBA is known to bind ubiquitylated Hpr1 (a component of THO/TREX) (*Gwizdek et al., 2006*). A competition of the UBA domain for binding to ubiquitylated Hpr1 and the internal Mex67 ubiquitylated site would provide a switch for Mex67 binding to THO/TREX.

Interestingly, Hpr1 ubiquitylation is mediated by Rsp5 (*Gwizdek et al., 2005*; *Gwizdek et al., 2006*). Hence, Rsp5-mediated ubiquitylation of Hpr1 and Mex67 deubiquitylation by Ubp15 would work hand in hand towards the assembly of Mex67 into mRNPs. This model, however, is at odds with our observation that *UBP15* deletion and catalytically inactive Ubp15 do revert Rsp5 export defects. Indeed, this genetic interaction suggests that the deletion of *UBP15* promotes export (i.e. the activity of Ubp15 would impair export), while the effect we observed on Mex67 interactions with THO and CBC would intuitively have the opposite effect (by promoting Mex67 interaction with THO and CBC, Ubp15 should have a positive effect on export). This conundrum may be solved by considering alternative scenarios. For instance, in *rsp5* mutant cells, Mex67 –which lost the ability to interact with THO/TREX via Hpr1– may (hypothetically) find another route towards assembly into mRNPs when ubiquitylated, a condition that would be favored by the deletion of *UBP15*. This would be reminiscent of what happens during heat-shock, where stress-response mRNAs are rapidly exported by

Mex67, without the need for adapters (*Zander et al., 2016*). This alternative mRNA export pathway allows fast export of stress response mRNAs but at the expense of mRNA quality control, which is bypassed. Assuming that *UBP15* deletion re-routes mRNA export toward such a pathway, one would expect *ubp15Δ* cells to accumulate aberrant transcripts in their cytoplasm, which we were not able to detect (data not shown). An alternative model, consistent with all our data, would be that Mex67 and Hpr1 ubiquitylation need to turnover for mRNA export to optimally function (see model in *Figure 5E*). Indeed, it sounds reasonable to think that Mex67 and Hpr1 need to flip back and forth between their ubiquitylated and non-ubiquitylated forms, considering the very dynamic nature of the interactions that they form and the fact that mRNPs are being remodeled during their journey from the transcription site to the cytoplasmic side of the nuclear envelope. For example, one may envision that deubiquitylation of Mex67 is important for its recruitment by THO but that its ubiquitylation is also important to allow Mex67 to let go of THO and escort the mRNA to the NPC. In summary, the exact molecular mechanisms describing the role of Ubp15 in mRNA export and its coupling to transcription will require additional work, but the data presented here clearly establish this deubiquitylase as a new player in this arena.

# Materials and methods

## Key resources table

| Reagent type (species) or resource | Designation | Source or reference | Identifiers |
|---|---|---|---|
| Strain, strain background (*S. cerevisiae*) | Various | This paper | NCBITaxon:4932 |
| Recombinant DNA reagent | Various | This paper | Plasmids |
| Antibody | anti-Flag M2 mouse monoclonal antibody | Sigma | Cat# F3165, RRID:AB_259529 |
| Antibody | anti-Rpb3 mouse monoclonal antibody (W0012) | Neoclone/Biolegend | Cat# 665003, RRID:AB_2564529 |
| Antibody | anti-HA F7 mouse monoclonal antibody | Santa Cruz Biotechnology | Cat# sc-7392, RRID:AB_627809 |
| Antibody | rabbit IgG | Sigma | Cat# I5006, RRID:AB_1163659 |
| Chemical compound, drug | Pan Mouse Dynabeads | Thermo Fisher Scientific | Cat# 11042 |
| Chemical compound, drug | Dynabeads M-270 Epoxy | Thermo Fisher Scientific | Cat# 14302D |
| Chemical compound, drug | Ni-NTA agarose beads | Qiagen | Cat# 30210 |
| Chemical compound, drug | mono-reactive NHS ester fluorescent Cy5 and Cy3 dyes | GE Healthcare | Cat# PA23001 Cat# PA25001 |
| Software, algorithm | Versatile Aggregate Profiler (version 1.1.0) | *Brunelle et al., 2015*; *Coulombe et al., 2014* | |

## Yeast strains and plasmids

Genotypes for the yeast strains used in this study are listed in *Supplementary file 3*. All tagged and deletion strains were done by homologous recombination of appropriate PCR cassettes. The catalytic dead *ubp15* mutation (C214A) was introduced into pFR559 by inverted PCR (forward: p-GCC TATTTGAATTCGTTATTGC, reverse: TGTGGCACCCTGATTTCGGAAGCCA) to generate pFR560 (see *Supplementary file 4*). Each construct was validated by sequencing and their expression tested by western blot analysis.

## Serial-dilution growth assays

Cells were grown to saturation in the indicated media at 30°C, washed, and resuspended to an OD$_{600}$ of 1.0 in H$_2$O. Cells were then subjected to fivefold serial dilutions and spotted onto the appropriate media. Plates were incubated at 30°C for 3 days unless specified otherwise. Images presented in figures are representative examples of at least two biological replicates, except for the

cold sensitivity (15°C) assay in *Figure 2—figure supplement 1B* and thermosensitivity assays in *Figure 4—figure supplement 1B–C*, which were done once. In addition, several phenotypes were confirmed in two different yeast genetic backgrounds.

## Purification of proteins associated with RNAPII, Ubp15, and Mex67

TAP-tagged Rpb1 subunit of RNAPII from WT and *fcp1-1* cells was purified in two biological replicates by one-step affinity purification essentially as published previously (*Jeronimo et al., 2015*). In brief, following cryogenic disruption of cells (*Trahan et al., 2016*), frozen cell grindate (5 g) was thawed into nine volumes of EB150 extraction buffer (20 mM Tris-HCl pH 7.5, 150 mM KOAc, 1 mM EDTA pH 8.0, 0.5% Triton X-100, 10% glycerol, 1 mM DTT, protease and phosphatase inhibitors, and 1:5000 Antifoam B [Sigma]), vortexed for 30 s and homogenized (Polytron PT 1200E; Kinematica AG) for another 30 s, to allow for maximal recovery of chromatin proteins. The cleared extract was incubated for 1 hr at 4 °C with 200 µL of pre-washed magnetic Dynabeads M-270 Epoxy (Thermo Fisher Scientific) conjugated to rabbit IgG (Sigma). Dynabeads were then collected and washed five times with EB150 buffer and two times with TEV150 protease cleavage buffer (10 mM Tris-HCl pH 8.0, 150 mM KOAc, 0.5 mM EDTA pH 8.0, 0.1% Triton X-100, 10% glycerol, and 1 mM DTT). The isolated protein complex was eluted by incubating the beads overnight at 4 °C with 200 units of TEV protease (Thermo Fisher Scientific) in 500 µL of TEV150 buffer. After digestion, the collected eluate was incubated with pre-washed nickel-nitrilotriacetic acid (Ni-NTA) agarose beads (Qiagen) for 90 min at 4°C to remove the His-tagged TEV protease. The collected eluate was concentrated to ~300 µl by dialysis in PEG dialysis buffer (10 mM HEPES-KOH pH 7.9, 0.1 mM EDTA pH 8.0, 100 mM KOAc, 20% glycerol, 20% PEG-8000 and 1 mM DTT) and then dialyzed in No-PEG dialysis buffer (10 mM HEPES-KOH pH 7.9, 0.1 mM EDTA pH 8.0, 100 mM KOAc, 20% glycerol and 1 mM DTT). A fraction of the purified proteins was separated by SDS-PAGE on a 4–12% NuPAGE Novex Bis-Tris precast gel (Thermo Fisher Scientific) and visualized by silver staining and western blot analysis with the indicated antibodies. Proteins were precipitated with TCA before mass spectrometry analysis.

For the purification of the Ubp15-TAP complex, the same one-step affinity purification procedure was used except that the salt concentration of the buffers involved in the preparation of the extract and the purification step was increased to 500 mM. In brief, EB500 extraction buffer (20 mM Tris-HCl pH 7.5, 500 mM KOAc, 1 mM EDTA pH 8.0, 0.5% Triton X-100, 10% glycerol, 1 mM DTT, protease and phosphatase inhibitors and 1:5000 Antifoam B [Sigma]) and TEV500 protease cleavage buffer (10 mM Tris-HCl pH 8.0, 500 mM KOAc, 0.5 mM EDTA pH 8.0, 0.1% Triton X-100, 10% glycerol and 1 mM DTT) were used.

For the purification of the proteins associated with Mex67-3Flag from WT and *ubp15Δ* cells, frozen cell grindates (1 g), from three biological replicates, were thawed into nine volumes of TBT extraction buffer (20 mM HEPES-KOH pH 7.5, 110 mM KOAc, 1 mM MgCl$_2$, 0.5% Triton X-100, 0.1% Tween 20, 1 mM DTT, protease inhibitor mixture, SUPERaseIn RNase inhibitor [Thermo Fisher Scientific] and 1:5000 Antifoam B [Sigma]) as previously described (*Oeffinger et al., 2007*). The cell extracts were then vortexed for 30 s, homogenized (Polytron PT 1200E; Kinematica AG) for another 30 s, and clarified by centrifugation at 3500 rpm for 10 min at 4°C. The Mex67-3Flag tagged protein was isolated using 200 µL of pre-washed magnetic Pan Mouse Dynabeads (Thermo Fisher Scientific) coupled to 34 µg of anti-Flag M2 mouse monoclonal antibody (Sigma). After binding for 1 hr at 4°C, Dynabeads were collected and washed five times with TBT extraction buffer and five times with TBT washing buffer (20 mM HEPES-KOH pH 7.5, 110 mM KOAc, 1 mM MgCl$_2$). The isolated protein complex was eluted twice by incubating each time the beads 20 min at room temperature with 500 µL of NH$_4$OH elution buffer (0.5 M NH$_4$OH, 1 mM EDTA pH 8.0). The pooled eluates were split into four aliquots, then dried in a speed-vac at room temperature. One aliquot of purified proteins was separated by SDS-PAGE on a 4–12% NuPAGE Novex Bis-Tris precast gel and visualized by silver staining. Another aliquot was analyzed by mass spectrometry.

## Protein identification by mass spectrometry

The experiments were essentially performed as described previously (*Bataille et al., 2012*). Protein samples were re-solubilized in 6 M urea buffer followed by reduction and alkylation before digestion with trypsin (Promega) at 37°C for 18 hr. The digested peptide mixtures were dried down in a vacuum centrifuge and stored at −20°C until LC-MS/MS analysis. Prior to LC-MS/MS, the digested

peptide mixtures were resolubilized in 0.2% formic acid and desalted/cleaned up by using C18 Zip-Tip pipette tips according to the manufacturer's instructions (Millipore). Eluates were dried down in a vacuum centrifuge and then re-solubilized in 2% ACN/1% formic acid. The LC column used was a C18 reversed-phase column packed with a high-pressure packing cell. A 75 µm i.d. Self-Pack PicoFrit fused silica capillary column of 15 cm long (New Objective) was packed with the C18 Jupiter 5 µm 300 Å reverse-phase material (Phenomenex). This column was installed on the Easy-nLC II system (Proxeon Biosystems) and coupled to the LTQ Orbitrap Velos or the Orbitrap Fusion (Thermo Fisher Scientific) equipped with a Proxeon nanoelectrospray ion source.

## Mass spectrometry data analysis

Protein database searching was performed with Mascot 2.2 or 2.6 (Matrix Science) against the *S. cerevisiae* NCBInr protein database (2010-12-14 release) or the UniProt_Saccharomyces_cerevisiae (559292 - strain ATCC 204508/S288 c) database. The mass tolerances for precursor and fragment ions were set to 15 ppm and 0.60 Da, respectively. Trypsin was used as the enzyme allowing for up to two missed cleavages. Carbamidomethyl and oxidation of methionine were allowed as variable modifications. Data interpretation was performed using Scaffold 3.1.2 or 4.8 (Proteome Software). Spectral counts values were exported in Excel and processed as follows.

For RNAPII purifications in WT and *fcp1-1* cells, spectral counts from two biological replicates of Rpb1-TAP purified from WT cells and two biological replicates of Rpb1-TAP purified from *fcp1-1* cells, together with a collection of no-tag controls, were used as a measure of protein abundance. Spectral counts for the 706 identified proteins were floored to 0.1 and normalized to the bait protein level (Rpb1). Proteins with an average spectral count in no-tag controls above 10 (n = 44) were removed. Proteins with less than five average spectral counts in both WT and *fcp1-1* conditions (n = 495) were also removed. The final dataset contains 170 proteins. For these, the $\log_2$ relative abundance ($\log_2$ (*fcp1-1*/WT)) and the average intensity (across all four samples) were calculated. These values are available in *Supplementary file 1* and displayed in *Figure 1B*.

For the Ubp15-TAP purification, spectral counts from Ubp15-TAP and a no-tag control were used as a measure of protein abundance. Proteins with more than 10 spectral counts in no-tag control (n = 91) were removed. A protein with less than twofold spectral count in the tag versus no-tag sample was also removed. Proteins with less than five spectral counts in the Ubp15-TAP sample (n = 126) were also removed. Finally, duplicated proteins (different IDs referring to the same protein) were removed (n = 3). This analysis led to a final dataset of 41 proteins displayed in *Figure 1C*.

For Mex67-3Flag purifications in WT and *ubp15Δ* cells, spectral counts from three biological replicates of Mex67-3Flag from WT, *ubp15Δ* and a no-tag control were used as a measure of protein abundance. Proteins with less than fivefold enrichment in tagged versus no-tag, and with an average of less than eight spectral counts in tagged experiments were removed. This analysis resulted in 47 proteins displayed in *Figure 5D*. The data were normalized to set the ratio of the spectral count of the bait (Mex67) to one between the WT and *ubp15Δ* samples. The $\log_2$ of the ratio between the normalized average spectral count in *ubp15Δ* and WT cells were computed and a t-test on the spectral counts of the three WT and three *ubp15Δ* samples was used to assess significance.

## ChIP-qPCR

To assess RNAPII (Rpb3) binding along the *YLR454W* gene, ChIP experiments were performed in two biological replicates as previously described (*Collin et al., 2019*). In brief, yeast cells containing the *GAL1* promoter controlling the *YLR454W* gene were inoculated from an overnight preculture in 100 mL of yeast nitrogen-based (YNB) medium lacking uracil (−URA) supplemented with 2% galactose and 2% raffinose (to induce *YLR454W* gene expression). Cells were grown at 30°C until $OD_{600}$ reaches 0.6–0.8 and then 50 mL were treated with 100 µg/mL 6AU for 30 min while the rest was left untreated. Cells were cross-linked with 1% formaldehyde for 30 min at room temperature and quenched with 125 mM glycine. Immunoprecipitation of RNAPII was done using 3 µL of Rpb3 antibody (W0012 from Neoclone) coupled to magnetic Pan Mouse Dynabeads (Thermo Fisher Scientific). ChIP DNA was analyzed by qPCR using primers targeting the 5' (forward: ACGCAAAGGAACTAGAGAACG, reverse: AATAGGACTCTCCGCCTTGTT) and 3' (forward: GGTCACAGATCTATTACTTGCCC, reverse: TTCAGGCTCCGTGTAGGAATTA) regions of the *YLR454W* open reading frame.

1% of each Input sample was analyzed in parallel and enrichments were expressed as a percent of Input using the following formula: $100*2^{[Ct_{Input}-6.644-Ct_{IP}]}$.

## ChIP-chip

ChIP-chip experiments from WT, $ubp15\Delta$, $dst1\Delta$, and $ubp15\Delta/dst1\Delta$ cells were performed in two biological replicates (except for the $ubp15\Delta/dst1\Delta$ strain in the absence of 6AU which was done only once) as previously described (*Collin et al., 2019*). In brief, yeast cells were inoculated from an overnight preculture in 50 mL of YNB−URA supplemented with 2% glucose. Cells were grown at 30℃ until $OD_{600}$ reaches 0.6–0.8 and treated (or not) with 100 µg/mL 6AU for 60 min. Cells were cross-linked with 1% formaldehyde for 30 min at room temperature and quenched with 125 mM glycine. Immunoprecipitation of RNAPII was done using 3 µL of Rpb3 antibody (W0012 from Neoclone) coupled to magnetic Pan Mouse Dynabeads (Thermo Fisher Scientific). ChIP and Input samples were labeled with mono-reactive NHS ester fluorescent Cy5 and Cy3 dyes (GE Healthcare), respectively, combined and hybridized for at least 18 hr on custom-designed Agilent microarrays containing 180,000 Tm-adjusted 60-mer probes covering the entire yeast genome with virtually no gaps between probes (*Jeronimo and Robert, 2014*). Microarrays were washed and scanned using an InnoScan900 (Innopsys) at 2 µm resolution.

## ChIP-chip data analysis

The ChIP-chip data were normalized using the Limma Loess method and replicates were combined as described previously (*Ren et al., 2000*). The data were subjected to one round of smoothing using a Gaussian sliding window with a standard deviation of 100 bp to generate data points in 10 bp intervals (*Guillemette et al., 2005*). Aggregate profiles were generated using the Versatile Aggregate Profiler (VAP) (*Brunelle et al., 2015*; *Coulombe et al., 2014*). In these analyses, only genes that are at least 1 kb long and with an average RNAPII enrichment $\log_2$ ratio over one were retained. Genes were virtually split in the middle and their 5′ and 3′ halves were aligned on the TSS and polyA site, respectively. The signal was then averaged in 10 bp bins. Coordinates of TSS and polyA sites are from *Xu et al., 2009*. Violin plots of RNAPII processivity were built by calculating the $\log_2$ ratio of Rpb3 occupancy in the last versus first 300 bp for each gene using VAP. The plots were generated using PlotsOfData (*Postma and Goedhart, 2019*).

## RNA FISH

RNA FISH was performed as described previously (*Babour et al., 2016*) with a few modifications. Briefly, cells were grown in 50 mL of YPD medium to an $OD_{600}$ of 0.6–0.8 and fixed with 4% paraformaldehyde for 45 min at room temperature on a rotating wheel. Cells were washed twice with cold phosphate buffer (100 mM $KHPO_4$ pH 6.4) and once with cold spheroplast buffer (phosphate buffer supplemented with 1.2 M sorbitol). Digestion of yeast cell wall was performed at 30℃ with 250 µg of zymolyase 100T (US Biological). Spheroplasts were carefully washed twice and resuspended in 1 mL of cold spheroplast buffer and 200 µL were attached to poly-L-lysine (Sigma)-coated coverslips for 30 min at room temperature. Unadhered cells were washed off and coverslips were stored in 70% ethanol at −20℃ for at least 2 hr. Hybridization was carried out in hybridization buffer (50% formamide, 10% dextran sulfate, 4× SSC, 1× Denhardts, 125 µg/mL *E. coli* tRNA, 500 µg/mL ssDNA, 10 mM ribonucleoside-vanadyl complex [NEB]) supplemented with 50 ng of Cy5-oligo-$dT_{45}$ probe for 12 hr at 37℃ in the dark. Coverslips were washed twice with 2× SSC at 37℃ for 15 min, once with 1× SSC at room temperature for 15 min and twice with 0.5× SSC at room temperature for 15 min, and finally with 1× PBS buffer containing 0.5 µg/mL 4′,6-diamidino-2-phenylindole (DAPI) and mounted onto ProLong Gold antifade reagent (Thermo Fisher Scientific) mounting media.

### Fluorescence microscopy

FISH images were acquired with a Retiga EXi aqua camera (mount on 0.70×) mounted on a DM5500B upright microscope (Leica) and magnified through a 100× oil immersion objective (NA = 1.3). Images of fluorescent probes were excited using an X-Cite Series 120Q light source (Lumen Dynamics) with the appropriate filters. All hardware parts were controlled with Volocity v5.0 software. Data presented in figures are representative fields of images from two biological replicates. For calculating the percentage of mRNA accumulation in the different strains, we manually

visualized 150 to 300 cells and counted the ones with visible polyA RNA accumulation in the nucleus.

## In vivo ubiquitylation assay

Ubiquitylated proteins were detected essentially as described previously (*Muratani et al., 2005*) with some modifications. Yeast strains transformed with a plasmid expressing polyhistidine-tagged ubiquitin (6His-Ub) under the control of a copper-inducible promoter (pFR453) or its non-tagged control (pFR452) (see *Supplementary file 4*) were grown in 50 mL of YNB−URA. When $OD_{600}$ reached 0.6–0.8, cells were induced with 500 µM $CuSO_4$ for 2–3 hr at 30°C. Cell cultures were centrifuged and washed twice with ice-cold water, flash-frozen, and stored at −80°C. Cell pellets were resuspended in 1 mL of freshly prepared A2 buffer (6 M guanidine-HCl, 100 mM $Na_2HPO_4$/$NaH_2PO_4$ pH 8.0, 10 mM imidazole, 250 mM NaCl, 0.5% NP40) and lysed by glass bead beating 5 min twice. Cell Lysates were then clarified by centrifuging for 15 min at max speed at 4°C. 10 µL of the cleared extract was kept as Input and the rest was used for Ni-NTA purification. His-tagged ubiquitin-conjugated proteins were purified by adding 100 µL of 50% Ni-NTA agarose beads (Qiagen) equilibrated in A2 buffer to 1 mL of cleared extracts and incubated with rotation for 2–4 hr at room temperature. Beads were pelleted and washed twice with 1 mL of A2 buffer, twice with 1 mL of A2/T2 buffer (1 vol A2 buffer + three volumes T2 buffer [50 mM $Na_2HPO_4$/$NaH_2PO_4$ pH 8.0, 20 mM imidazole, 250 mM NaCl, 0.5% NP40]) and twice with 1 mL of T2 buffer. Samples were rotated 5 min at room temperature for each wash. After the final wash, all liquid was removed from beads, and beads were resuspended in 50 µL of 2× Laemmli buffer supplemented with 250 mM imidazole. Inputs were prepared as previously described (*Pepinsky, 1991*). Briefly, 800 µL of 100% ethanol was added to 10 µL of cleared extract, vortexed, and incubated for 1 hr at −20°C. Inputs were then centrifugated and washed twice with 100% ethanol for 15 min at −20°C. Ethanol was removed and, when completely dry, pellets were resuspended in 100 µL of 2× Laemmli buffer. All samples were boiled for 5 min, centrifuged and supernatants were run on SDS-PAGE for western blot analysis with anti-HA F7 mouse monoclonal antibody (Santa Cruz Biotechnology). Images presented in figures are representative examples of at least three (*Figure 5*) or two (*Figure 5—figure supplement 1*) biological replicates, except for Nup133, which was assayed only once.

## Acknowledgements

This work was funded by grants from the Natural Sciences and Engineering Research Council of Canada (NSERC) to FR and the Canadian Institutes of Health Research (CIHR) to FR (MOP-162334) and JC (FDN-143314). FR holds a Research Chair from the « Fonds de Recherche Québec –Santé » (FRQS). JC holds the Canada Research Chair in Chromatin Biology and Molecular Epigenetics. FE held fellowships from « Fondation pour la Recherche Médicale » and FRQS. We are grateful to F Bachand, M Oeffinger, and D Zenkluzen for critical reading of the manuscript and C Dargemont for helpful discussions. We thank P Bensidoun and A Babour for insights about FISH experiments. We also thank A Fradet-Turcotte, B Coulombe, D Finley, M Kobor, D Stillman, K Struhl, F Winston, P Hieter, and JM Huibregtse for sharing reagents.

## Additional information

### Funding

| Funder | Grant reference number | Author |
| --- | --- | --- |
| Natural Sciences and Engineering Research Council of Canada | 435833-2013 | François Robert |
| Canadian Institutes of Health Research | MOP-162334 | François Robert |
| Canadian Institutes of Health Research | FDN-143314 | Jacques Cote |

The funders had no role in study design, data collection and interpretation, or the decision to submit the work for publication.

## Author contributions

Fanny Eyboulet, Conceptualization, Investigation, Methodology, Writing - original draft, Writing - review and editing; Célia Jeronimo, Conceptualization, Investigation, Methodology, Writing - review and editing; Jacques Côté, Supervision, Funding acquisition, Writing - review and editing; François Robert, Conceptualization, Formal analysis, Supervision, Funding acquisition, Writing - original draft, Project administration, Writing - review and editing

## Author ORCIDs

Jacques Côté (iD) https://orcid.org/0000-0001-6751-555X
François Robert (iD) https://orcid.org/0000-0003-1054-5920

## Decision letter and Author response

Decision letter https://doi.org/10.7554/eLife.61264.sa1
Author response https://doi.org/10.7554/eLife.61264.sa2

# Additional files

## Supplementary files

• Supplementary file 1. List of the proteins associated with RNAPII and their differential association in *fcp1-1* cells.

• Supplementary file 2. List of the proteins associated with Mex67 and their differential association in *ubp15Δ* cells.

• Supplementary file 3. List of yeast strains used in this study.

• Supplementary file 4. List of plasmids used in this study.

• Transparent reporting form

## Data availability

Microarray data and processed files have been deposited in GEO under the accession number GSE154671. Mass spectrometry data have been deposited in MassIVE under accession numbers MSV000085729, MSV000085730 and MSV000085731.

The following datasets were generated:

| Author(s) | Year | Dataset title | Dataset URL | Database and Identifier |
|---|---|---|---|---|
| Eyboulet F, Jeronimo C, Cote J, Robert F | 2020 | The Deubiquitylase Ubp15 Couples Transcription to mRNA Export | https://www.ncbi.nlm.nih.gov/geo/query/acc.cgi?acc=GSE154671 | NCBI Gene Expression Omnibus, GSE154671 |
| Eyboulet F, Jeronimo C, Cote J, Robert F | 2020 | The Deubiquitylase Ubp15 Couples Transcription to mRNA Export | https://massive.ucsd.edu/ProteoSAFe/dataset.jsp?task=3141be92-f8904a4eafd7906576-d4659f | Mass Spectrometry Interactive Virtual Environment, MSV000085729 |
| Eyboulet F, Jeronimo C, Cote J, Robert F | 2020 | The Deubiquitylase Ubp15 Couples Transcription to mRNA Export | https://massive.ucsd.edu/ProteoSAFe/dataset.jsp?task=df13595b88214e4a-b22ad4f0b92a0e77 | Mass Spectrometry Interactive Virtual Environment, MSV000085730 |
| Eyboulet F, Jeronimo C, Cote J, Robert F | 2020 | The Deubiquitylase Ubp15 Couples Transcription to mRNA Export | https://massive.ucsd.edu/ProteoSAFe/dataset.jsp?task=8a68c4aa270a49b7bf67-f7b1366ab00c | Mass Spectrometry Interactive Virtual Environment, MSV000085731 |

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
