## [Decision Letter]

**Acceptance summary:**

This manuscript uses a powerful combination of proteomics, genomics, biochemistry and genetics to identify Ubp15 as an RNA pol. II interacting protein that helps assemble mRNPs by deubiquitination of the major nuclear export receptor Mex67. This is the first ubiquitin protease shown to function in nuclear export.

**Decision letter after peer review:**

Thank you for submitting your article "The Deubiquitylase Ubp15 Couples Transcription to mRNA Export" for consideration by *eLife*. Your article has been reviewed by three peer reviewers, including Jerry L Workman as the Reviewing Editor and Reviewer #1, and the evaluation has been overseen by James Manley as the Senior Editor.

The reviewers have discussed the reviews with one another and the Reviewing Editor has drafted this decision to help you prepare a revised submission.

Summary:

The paper by Eyboulet et al. identifies a new role for the ubiquitin protease Ubp15 in coupling transcription and mRNA export. Briefly, the authors first show that Ubp15 interacts with hyperphosphorylated RNA PolII suggesting a role in transcription elongation (Figure 1). Interestingly, loss of Ubp15 suppresses the 6AU sensitivity of dst1Δ and several other elongation mutants (Figure 2). Unexpectedly however, loss of Ubp15 does not rescue the RNA PolII processivity defect of dst1Δ, suggesting that ubp15Δ suppresses the phenotype more indirectly (Figure 3). The authors then identify Rsp5 as an E3 ligase opposing the effect of Ubp15 and show that the mRNA export defect observed in the rsp5-1 mutant is suppressed in the absence of Ubp15 (Figure 4). Furthermore, they show that the export receptor Mex67 is ubiquitinated by Rsp5 and that ubiquitination of Mex67 (but not of several other NUPs or mRNA export factors) is increased in the absence of Ubp15, indicating that Rsp5 and Ubp15 have opposing effects on Mex67 (Figure 5A, B, C). Finally, performing mass spec analyses of Mex67 interacting proteins in WT vs ubp15Δ, they find that increased Mex67 ubiquitination in the absence of Ubp15 decreases the interaction of the export receptor with components of the Cap binding and THO complexes involved in the recruitment of Mex67 to nascent transcripts (Figure 5D).

1) Mechanistically, Eyboulet et al. shows many interesting results regarding novel finding of Ubp15 in mRNA export. The only problem is how ubiquitylation and deubiqutylation cycle can regulate Mex67-mediated mRNA export process. Especially, rescue phenotype of double mutants of Rsp5 and UBP15 seems to be counterintuitive. More detailed mechanistic approaches will be required to explain the function of Ubp15, which may be beyond the scope here. However, do the authors have this data or information of Mex67 occupancy in the absence of those mutants? Alternatively, they can discuss their model to explain how Ubp15 regulates Mex67 using current view about mRNA export.

2) The authors should check Rsp5 protein levels in rsp5-1 ts mutant in the absence of UBP15. Sometimes, protein expression can be rescued by certain deletion mutants and thus they may see rescue phenotype.

3) Mex67 distribution in the absence of UBP15 and/or RSP5 ts on chromatin. Does ubiquitylation cycle change its distribution (or any delay) on chromatin?

4) From Figure 4 and 5 and related to #1 comment, absence of UBP15 alone did not show any defect in mRNA export but did deregulate the interaction of Mex67 with THO and the cap-binding complex. If ubiquitylation of Mex67 is important for downstream pathway of mRNA export, ubp15D should show mRNA export defect. How do you explain this?

Revisions expected in follow-up work:

How ubiquitylation and deubiqutylation cycle can regulate Mex67-mediated mRNA export process.

---

## [Author Response]

Revisions for this paper:1) Mechanistically, Eyboulet et al. shows many interesting results regarding novel finding of Ubp15 in mRNA export. The only problem is how ubiquitylation and deubiqutylation cycle can regulate Mex67-mediated mRNA export process. Especially, rescue phenotype of double mutants of Rsp5 and UBP15 seems to be counterintuitive. More detailed mechanistic approaches will be required to explain the function of Ubp15, which may be beyond the scope here. However, do the authors have this data or information of Mex67 occupancy in the absence of those mutants? Alternatively, they can discuss their model to explain how Ubp15 regulates Mex67 using current view about mRNA export.

As we mentioned above, we indeed have performed ChIP-chip experiments for Mex67 (together with RNAPII as a control) in WT and *ubp15Δ* cells (see Author response image 1). Based on our proteomic analysis of Mex67 in *ubp15Δ* cells, our (naïve) expectation was that *UBP15* deletion would lead to decreased association of Mex67 with transcribed genes. Unfortunately, we observed no convincing effect of *ubp15Δ* on Mex67 ChIP patterns. This negative result, however, needs to be interpreted with caution since Mex67 ChIP leads to mediocre enrichment over background. In that context, we do not exclude the possibility that *UBP15* deletion affects Mex67 recruitment to genes but that the effect is too subtle to be detected in the context of a low signal-to-noise. Alternatively, *ubp15Δ* may affect a population of Mex67 molecules that is not detected by ChIP. For example, Ubp15 may regulate the transfer of Mex67 from THO to the mRNA. These ChIP experiments are now mentioned in the Discussion as “data not shown”, together with an enhanced discussion of possible mechanisms. Author response image 1 could be integrated as a supplement to Figure 5, should you think it would improve the manuscript.

**Author response image 1. sa2fig1:** Mex67 occupancy on active genes is not affected by the deletion of *UBP15*. A) RNAPII (Rpb3) occupancy, as determined by ChIP-chip, over genes longer than 1 kb and with an average RNAPII occupancy (log_2_ ratio) over 1 (n=234), in WT and *ubp15Δ* cells. B) Mex67-3xFLAG occupancy, as determined by ChIP-chip, over genes longer than 1 kb and with an average RNAPII occupancy (log_2_ ratio) over 1 (n=234), in WT and *ubp15Δ* cells. The left panel displays the data using the same y-axis as panel A, while the right panel shows a zoom into the -0.3-0.2 log_2_ ratio range.

2) The authors should check Rsp5 protein levels in rsp5-1 ts mutant in the absence of UBP15. Sometimes, protein expression can be rescued by certain deletion mutants and thus they may see rescue phenotype.

This is a very good point. We have performed this experiment and found that Rps5-1 protein levels actually decrease upon deletion of *UBP15*. We also overexpressed Rsp5-1 (by placing it under the *ADH1* promoter) and found that this suppresses the ts phenotype. Hence, the *rsp5-1* ts phenotype can be suppressed by overexpression of the Rsp5-1 protein but this protein is actually downregulated upon deletion of *UBP15*. Taken together, these data show that the mechanism through which the deletion of *UBP15* suppresses the *rsp5-1* phenotype cannot be via increasing Rsp5 protein levels. These data are now shown in Figure 4—figure supplement 1C-D.

3) Mex67 distribution in the absence of UBP15 and/or RSP5 ts on chromatin. Does ubiquitylation cycle change its distribution (or any delay) on chromatin?

See our response to point 1 and Author response image 1.

4) From Figure 4 and 5 and related to #1 comment, absence of UBP15 alone did not show any defect in mRNA export but did deregulate the interaction of Mex67 with THO and the cap-binding complex. If ubiquitylation of Mex67 is important for downstream pathway of mRNA export, ubp15D should show mRNA export defect. How do you explain this?

There is a conundrum here indeed, and we have spent a lot of energy trying to solve it. At this point, we can only speculate, and this is part of the Discussion section. In sum, here is what we think: The fact that *UBP15* deletion rescues the defect of *rsp5-1* cells suggests that the deletion of *UBP15* somehow makes export more efficient (at least in the context of *rsp5-1*). Assuming this is the case, it is not surprising that *UBP15* deletion does not lead to mRNA export defect on its own (the expected phenotype would be excess export, which would not translate into a change in FISH profiles). Considering that the deletion of *UBP15* makes export more efficient, however, is not easily reconciled with our proteomics data showing that *UBP15* deletion leads to reduced interactions between Mex67 and THO/CBC. Intuitively, reducing these interactions should impair export. One possible scenario would be that Ubp15 plays a role in mRNA quality control. In this context, by uncoupling Mex67 from THO and CBC, the deletion of *UBP15* would allow Mex67 to bypass its normal path towards its loading on the mRNA (which is part of how mRNA are quality controlled). In this model, preventing Mex67 to interact with THO/CBC would not lead to reduced export, but rather to a bypass of the quality control checkpoint. A similar phenomenon was shown to occur during heat-shock where heat-shock mRNAs are rapidly exported in association with Mex67 but without the need for adapters (that is, without being recruited via THO) (PMID: 27951587). That model predicts that *ubp15Δ* cells may show increased levels of aberrant transcripts in their cytoplasm. We therefore used two previously described assays to look at this phenomenon. First, we used the so-called “Leakage assay” (PMID: 2655924, 9130720) that measures the leakage of unspliced transcript in the cytoplasm. Second, we used the *lys2-370* mutant which, despite coding for a functional Lys2 protein, does not support growth on -LYS media because the *lys2-370* mRNA is recognized as aberrant and not exported. Impairing mRNA quality control is known to rescue this growth defect (PMID: 16832048, Barbour et al., 2016). Unfortunately, neither of these assays provided evidence for a defect in mRNA quality control in *ubp15Δ* cells (Author response image 2). We are therefore left with more complicated models that would require a considerable amount of work to test. One possibility is that the dynamics of the ubiquitylation/deubiquitylation cycle is key, so that the double mutant re-establishes some equilibrium relative to what happens in *rsp5-1* cells. All these scenarios are discussed in the Discussion section.

**Author response image 2. sa2fig2:** Deletion of *UBP15* does not lead to increased levels of aberrant mRNAs in the cytoplasm. A) % of intron-containing mRNAs leaking into the cytoplasm, as measured using the “Leakage assay” described before (Legrain and Rosbash, 1989, Rain and Legrain, 1997) in WT, *ubp15Δ* and *mlp1Δ* cells. *mpl1Δ* is used as a positive control. B) The indicated mutants were grown overnight in YPD, serial-dilated (10-fold), and plated on YMB containing a complete set of amino acids (complete) and YNB lacking lysine (-LYS) Growth on – LYS indicates that the defective (but still functional) *lys2-370* transcript is exported in the cytoplasm. *rrp6Δ* is used as a positive control.

Revisions expected in follow-up work:How ubiquitylation and deubiqutylation cycle can regulate Mex67-mediated mRNA export process.

As discussed in length above, the revised manuscript now includes an enhanced discussion of this outstanding question in the Discussion section. We notably discuss additional data (ChIP-chip and mRNA export quality control assays). These data are currently not shown as a figure but could be added as figure supplements should you think it is the best approach.